# Growth of Wide-Bandgap Monolayer Molybdenum Disulfide for a Highly Sensitive Micro-Displacement Sensor

**DOI:** 10.3390/nano14030275

**Published:** 2024-01-27

**Authors:** Shaopeng Wang, Jiahai Huang, Yizhang Wu, Huimin Hao

**Affiliations:** 1College of Mechanical and Vehicle Engineering, Taiyuan University of Technology, Taiyuan 030024, China; wsp19961026@163.com (S.W.); huangjiahai@tyut.edu.cn (J.H.); 2College of Science, Hohai University, Nanjing 211100, China; yizhwu@unc.edu; 3Department of Applied Physical Sciences, The University of North Carolina at Chapel Hill, Chapel Hill, NC 25714, USA

**Keywords:** 2D wide-bandgap semiconductor material, MoS_2_, piezoelectric effect, chemical vapor deposition, micro-displacement sensor

## Abstract

Two-dimensional (2D) piezoelectric semiconductor materials are garnering significant attention in applications such as intelligent sensing and energy harvesting due to their exceptional physical and chemical properties. Among these, molybdenum disulfide (MoS_2_), a 2D wide-bandgap semiconductor, exhibits piezoelectricity in odd-layered structures due to the absence of an inversion symmetry center. In this study, we present a straightforward chemical vapor deposition (CVD) technique to synthesize monolayer MoS_2_ on a Si/SiO_2_ substrate, achieving a lateral size of approximately 50 µm. Second-harmonic generation (SHG) characterization confirms the non-centrosymmetric crystal structure of the wide-bandgap MoS_2_, indicative of its piezoelectric properties. We successfully transferred the triangular MoS_2_ to a polyethylene terephthalate (PET) flexible substrate using a wet-transfer method and developed a wide-bandgap MoS_2_-based micro-displacement sensor employing maskless lithography and hot evaporation techniques. Our testing revealed a piezoelectric response current of 5.12 nA in the sensor under a strain of 0.003% along the armchair direction of the monolayer MoS_2_. Furthermore, the sensor exhibited a near-linear relationship between the piezoelectric response current and the strain within a displacement range of 40–100 µm, with a calculated response sensitivity of 1.154 µA/%. This research introduces a novel micro-displacement sensor, offering potential for advanced surface texture sensing in various applications.

## 1. Introduction

With the rapid development of modern science and technology, intelligent robots gradually occupy an increasingly important position in our lives. Accordingly, as the “skin” of robots, tactile sensors, of which the micro-displacement sensor is a fundamental module which has also been widely used in intelligent sensing, have been widely studied. Existing micro-displacement sensors are mainly piezoresistive, capacitive, piezoelectric, etc. [1]. Among them, the piezoresistive micro-displacement sensor reflects fluctuation in external stress based on the piezoresistive effect [2,3,4,5,6,7]. The capacitive micro-displacement sensor changes the capacitance of the sensitive element by altering the capacitive pad spacing caused by external stress and then reflects the texture information of the surface of the measured object [8,9,10,11,12]. Based on their working principles, piezoresistive and capacitive sensors have restricted detection limits and complex device designs.

The piezoelectric micro-displacement sensor detects the change in external stress via the piezoelectric effect of the sensitive element. When the sensitive element with the piezoelectric effect is subjected to external stress, piezoelectric potential is generated inside it so that the external force is reflected by the change in the output current/voltage. The detection range and sensitivity of these sensors are higher than those of the piezoresistive and capacitive sensors. Among the sensitive elements of the sensor, piezoelectric polyvinylidene fluoride (PVDF) is currently the most common piezoelectric material used for tactile perception. Compared with piezoelectric ceramics [13], PVDF has a good mechanical flexibility and can be used to prepare flexible tactile sensors [14,15,16,17,18,19]. However, the piezoelectric response of PVDF is unstable, and its sensitivity is not sufficient. Generally, it is necessary to design a microstructure to improve the sensor’s detection ability, so the device structure is also more complex. As a typical representative of transition metal dihalides, monolayer molybdenum disulfide has piezoelectric characteristics [20,21,22,23], a wide bandgap, and good flexibility and mechanical strength and has been widely used in instrument science, electronics, optics, and many other fields.

In this paper, a wide-bandgap monolayer MoS_2_-based micro-displacement sensor has been proposed. Firstly, the monolayer MoS_2_ was prepared on a silicon oxide substrate using the chemical vapor deposition (CVD) method and was then transferred to a polyethylene terephthalate (PET) flexible substrate via the wet-transfer method. The electrodes were prepared on the surface of the MoS_2_ to complete the preparation of the micro-displacement sensor. A micro-displacement testing system was used to test the sensor. The results showed that the piezoelectric-induced current of the sensor had an approximate linear relationship with the strain applied to the MoS_2_ along the armchair direction. The calculated response sensitivity of the device was 1.154 µA/%, which is expected to perceive fine textures on the surfaces of objects.

## 2. Preparation of the Micro-Displacement Sensor

### 2.1. Preparation of the Sensitive Element (Wide-Bandgap Monolayer MoS_2_)

The monolayer MoS_2_ was prepared using the CVD method, as shown in Figure 1. Compared with other methods such as stripping and the hydrothermal method, the CVD method can prepare two-dimensional materials with a large size, a regular shape, and a controllable layer number, and it has good repeatability. Figure 1 shows the diagram of the principle behind the preparation of single-layer molybdenum disulfide using the CVD method. A SiO_2_/Si substrate was selected as the growth substrate of monolayer molybdenum disulfide because SiO_2_ has the same hexagonal lattice structure as single-layer molybdenum disulfide, thus ensuring lattice matching between the substrate and the material, which was conducive to the growth of molybdenum disulfide. In the experiment, ammonium molybdate tetrahydrate and elemental sulfur were selected as the precursors of the molybdenum source and the sulfur source for CVD growth. The aqueous solution of ammonium molybdate tetrahydrate was dropped on one end of the SiO_2_/Si substrate and dried. The solution was placed in the middle of the tube furnace, and the elemental sulfur was placed upstream position of the tube furnace. After reaching the target temperature, the ammonium molybdate tetrahydrate in the center of the tube furnace was heated to decompose the molybdenum atoms, and then the elemental sulfur was heated to evaporate into gaseous sulfur atoms and react with the molybdenum atoms to form molybdenum disulfide molecules under the transportation of argon. Finally, the molybdenum disulfide nucleated and grew on the SiO_2_/Si substrate. The whole growth process can be divided into three stages: (1) ammonium molybdate tetrahydrate is decomposed to obtain molybdenum atoms due to the increase in temperature; (2) elemental sulfur is heated to evaporate into gas and reaches the central region under the transport of argon to react with the molybdenum atoms and form molybdenum disulfide molecules; (3) the molybdenum disulfide molecules nucleate on the SiO_2_/Si substrate and grow into triangular single-crystal films. It is worth noting that the nucleation of molybdenum disulfide on the substrate is crucial. The nucleation of two-dimensional materials is usually easy to achieve in the defects of a substrate, while the SiO_2_/Si substrate’s surface is smooth and clean, with few defects, meaning that it is not easy to nucleate molybdenum disulfide on this substrate. Therefore, in this experiment, we spin-coated a layer of sodium cholate aqueous solution on the surface of the SiO_2_/Si substrate and dried it. The nucleation promotion point was artificially made to facilitate the nucleation of single-layer molybdenum disulfide.

In a Thermo Scientific single-temperature zone tube furnace, a silicon (Si) substrate (Harbin Ultra-Thin Technology Co., Ltd., Harbin, China) with 300 nm silicon dioxide (SiO_2_) was placed in the center of the quartz tube in the tube furnace; the sulfur source was placed near the upstream edge of the furnace body in the tube furnace; and monolayer MoS_2_ was grown with argon as the carrier gas. A sodium cholate aqueous solution (10 g deionized water + 100 mg sodium cholate powder (SIGMA ALDRICH, Saint Louis, MO, USA) was configured as the nucleation promoter for molybdenum disulfide growth, and a saturated ammonium molybdate (SIGMA ALDRICH, USA) aqueous solution was configured as the molybdenum source. Compared to the typical molybdenum source precursor of MoO_3_, ammonium molybdate can be soluble in water and can reach saturation in an ammonium molybdate aqueous solution. The aqueous solution can precisely control the amount of molybdenum source precursor used [24,25,26,27]. Pure elemental sulfur (SIGMA ALDRICH) was used as the sulfur source in our study. The temperature of the molybdenum source in the center of the tube furnace was 680 °C; the heating rate of the tube furnace was 30 °C/min; the holding time was 20 min; the carrier gas flow was 580 sccm; and the temperature was cooled to room temperature naturally after the end of the heating process.

### 2.2. Transfer of Wide-Bandgap Monolayer MoS_2_ on a Si Substrate to a PET Flexible Substrate

At present, the transfer methods for two-dimensional materials mainly include dry-transfer and wet-transfer methods [28,29]. Among them, the wet-transfer method mainly relies on chemical etching to achieve the transfer of materials, and this transfer process is shown in Figure 2. Firstly, polymethyl methacrylate (PMMA) was spin-coated on the surface of the Si/SiO_2_ substrate and then 2 mol/L potassium hydroxide (KOH) solution was used as the etching agent for removing the SiO_2_. The Si/SiO_2_ substrate was soaked in a KOH solution for 1 h to bind MoS_2_ to the surface of PMMA, and the PMMA attached to the MoS_2_ was transferred to the surface of PET. The PMMA was then etched in acetone, and the monolayer MoS_2_ was finally transferred to the PET flexible substrate.

### 2.3. Preparation of Electrodes

A pair of electrodes were deposited on the surface of MoS_2_ in the armchair direction. The procedure of electrode preparation included photolithography and evaporation, as shown in Figure 3. The photolithography process mainly includes etching patterned grooves on the photoresist material through exposure, development, and other operations so as to facilitate the deposition of subsequent electrodes. According to the requirements of a mask or not, lithography can be divided into masked lithography and mask-free lithography. In the experiment, a HELDELBERG INSTRUMENTS μMLA maskless lithography machine was used. Maskless lithography has the advantages of a simple operation and a high precision and can realize the patterning operation on a nanometer scale. An AZ150Q photoresist (Suzhou JUXINMEMS Technology Co., Ltd, Suzhou, China) and an AZ300mif developer (Suzhou JUXINMEMS Technology Co., Ltd, Suzhou, China) were selected to develop the pattern. The laser’s focusing and energy parameters were 60 and 0, respectively. The chromium/gold electrode was steamed on the surface of the MoS_2_ on the PET substrate using a composite coating system of thermal evaporation and electron beam evaporation, with a thickness of 5 nm for chromium and 50 nm for gold. After the end of the lithography process, the surface of the PET substrate was covered with photoresist material except for the exposed electrode pattern area. After the evaporation of the electrode, the substrate was soaked in an acetone solution, and the acetone dissolved all the photoresist on the substrate. So, except for the electrode pattern area, the Cr/Au vaporized in other areas of the substrate was stripped off of the substrate with the photoresist. Finally, we could successfully vaporize the electrode.

## 3. Test Principle of Micro-Displacement Sensor

The test principle of the micro-displacement sensor is shown in Figure 4. Here, we use a cantilever beam with one end fixed and the other end free to test the piezoelectric response of the molybdenum disulfide micro-displacement sensor. The thickness, length, and width of the PET substrate are a, l, and w, respectively. One end of the PET substrate is fixed on the base, and the other end is flush with the drive rod of the linear motor to keep a state of free movement. The molybdenum disulfide micro-displacement sensor is on the surface of the PET substrate, and its distance from the fixed end is z. During the test, the linear motor makes the free end of the PET substrate move laterally by pushing the drive rod, and the movement displacement is D_max_. The PET substrate is bent under the push of the driving rod. In this state, the monolayer MoS_2_ on the surface of the substrate is in a tensile state along the armchair direction. The tensile strain of the monolayer MoS_2_ can be calculated using the cantilever model of the PET substrate and Saint-Venant’s small deformation theory. Monolayer MoS_2_ possesses a positive triangular prism non-centrosymmetric crystal structure in which molybdenum atoms are located in the center, and sulfur atoms are distributed in the six vertices. When subjected to external stress along the armchair direction of MoS_2_, the charge center of the cation and anion within the molecule undergoes a relative displacement, resulting in a dipole moment and potential distribution along the direction of the stress. This shows the piezoelectric effect. When the external force disappears, the piezoelectric potential also disappears.

According to Figure 4, the mechanical equilibrium equation under the condition that the sensor is fixed below and free above is as follows [30]: (1)$$\nabla \cdot \sigma ={\vec{f}}_{e}^{\left(b\right)}=0$$
where ${\vec{f}}_{e}^{\left(b\right)}$ is the force applied to the material; ∇ is the divergence operator; and σ represents the stress tensor, which is associated with strain *ε*, electric field *E*, and electric displacement *D* by the constitutive equation below [31].
(2)$$\left\{\begin{array}{c} \sigma_{p}=c_{pq}\varepsilon_{q}-e_{kp}E_{k}\\ D_{i}=e_{iq}\varepsilon_{q}+k_{ik}E_{k} \end{array}\right.$$
where $c_{pq}$ is the linear elastic constant; $e_{kp}$ and $e_{iq}$ are the linear piezoelectric coefficients with a mutual relationship; $k_{ik}$ is the dielectric constant; $E_{k}$ is the electric field; and $D_{i}$ is the electric displacement. Monolayer MoS_2_ belongs to the $D_{3h}$ point group, so its linear elastic coefficient, piezoelectric coefficient, and dielectric constant can be written as follows [32]:$$c_{pq}=\left[\begin{array}{cccccc} c_{11} & c_{12} & 0 & 0 & 0 & 0\\ c_{12} & c_{11} & 0 & 0 & 0 & 0\\ 0 & 0 & \frac{c_{11}-c_{12}}{2} & 0 & 0 & 0\\ 0 & 0 & 0 & 0 & 0 & 0\\ 0 & 0 & 0 & 0 & 0 & 0\\ 0 & 0 & 0 & 0 & 0 & 0 \end{array}\right]e_{ij}=\left[\begin{array}{cccccc} e_{11} & -e_{11} & 0 & 0 & 0 & 0\\ 0 & 0 & 0 & 0 & 0 & -e_{11}\\ 0 & 0 & 0 & 0 & 0 & 0 \end{array}\right]$$

Assuming that there is no free charge in the piezoelectric material, the Gaussian equation and the geometric compatibility equation are satisfied, as follows [33]:(3)$$\nabla \cdot D=\rho_{e}^{\left(b\right)}=0$$
(4)∇×∇×ε=0

One end of the PET is fixed on the base, and the other end is free. The PET substrate can be regarded as a cantilever beam structure and the device on the PET surface can be connected with the external circuit to form a closed loop. The linear motor drives the push rod to apply a set displacement to the free end of the substrate, and the substrate is therefore bent. According to the beam structure theory and Saint-Venant’s small deformation theory, we can calculate the strain of the device from the displacement of the free end of the substrate.

As shown in Figure 4, the PET substrate can be approximated as a beam structure with length l, width w, and thickness a. The coordinate origin O is defined as the center of the fixed end, and the x and z axes are along the width and length directions of the substrate, respectively. Compared with the PET substrate, the size of MoS_2_ is very small, and the device is on the surface of the substrate. We can assume that MoS_2_ will be subjected to a pure tensile/compressive strain when the substrate is bent, which can be considered as the $\varepsilon_{zz}$ strain component of the PET substrate according to Saint-Venant’s small deformation theory [34]: (5)$$\sigma_{zz}=-(\frac{f_{y}}{I_{xx}})y(l-z)$$
where $I_{xx}$ is the moment of inertia of the beam structure, and $f_{y}$ is the force exerted by the push rod on the free end of the substrate. Compared to $f_{y}$ exerted by the push rod, it is easier to measure the displacement $D_{max}$ of the free end of the substrate. According to the beam structure theory [35],
(6)$$D_{max}=f_{y}l^{3}/3EI_{xx}$$
where *E* is Young’s modulus of the PET substrate. Accordingly, we have the following:(7)$$\sigma_{zz}=-(\frac{3ED_{max}}{l^{3}})y(l-z)$$
(8)$$\varepsilon_{zz}=\frac{\sigma_{zz}}{E}=-(\frac{3D_{max}}{l^{3}})y(l-z)$$

Here, the coordinates of MoS_2_ along the thickness direction and the length direction are y = ±a/2, z = z_0_, where z_0_ is the distance between MoS_2_ and the fixed end of the substrate. Therefore, the strain of MoS_2_ is as follows:(9)$$\varepsilon =\varepsilon_{zz}=\mp (\frac{3aD_{max}}{2l^{3}})(l-z_{0})$$

By substituting Formulas (7) and (9) into Formula (2), the electric field strength of the device along the armchair direction of MoS_2_ can be obtained as follows:(10)$$E_{k}=\frac{c_{pq}-E}{e_{kp}}$$

## 4. Characterization and Performance Testing of the Micro-Displacement Sensor

### 4.1. Characterization of Wide-Bandgap Monolayer MoS_2_

The obtained wide-bandgap monolayer MoS_2_ is shown in Figure 5. Optical microscopy is the basic means for characterizing two-dimensional materials, a technique through which we can clearly observe the surface morphology and size of a sample. The optical microscope used in this experiment was the Nikon ECLIPSE LV100ND (Nikon Precision Machinery Co., Ltd, Shanghai, China) optical microscope. Figure 5a is the optical microscopy image of the prepared wide-bandgap MoS_2_, in which the MoS_2_ samples are randomly distributed on the Si/SiO_2_ substrate, and the lateral size of the samples is approximately 50 µm. Some of the molybdenum disulfide samples were darker in color and not regular triangles in shape. The individual triangular samples measured more than 50 microns in their lateral size, and the surface morphology of the samples was uniform. We could initially judge the thickness of the samples by their optical contrast under the microscope. The thicker samples showed a darker color, while the thinner samples were lighter. Scanning electron microscopy (SEM) is another important means to characterize the morphology of two-dimensional materials, and the surface morphology of our samples could be clearly observed through SEM imaging. SEM was used to characterize one of the samples in our experiment. As shown in Figure 5b, the sample was triangular in shape, with a side length of about 50 microns. The surface of the sample was clean, without granular nucleation points, and the surface was flat and uniform. Raman spectroscopy is a kind of accurate, rapid, and non-destructive detection method for finding the composition and molecular structure of substances using the Raman scattering effect. As shown in Figure 5c, Raman spectroscopy of the obtained wide-bandgap MoS_2_ was performed at room temperature using a 532 nm laser line, which showed two peaks at 386 cm^−1^ and 403 cm^−1^, corresponding to the two molecular oscillations E_2g_^1^ and A_1g_ of monolayer MoS_2_ reported in the literature, respectively [36]. Meanwhile, the wave-number difference between the two peaks was less than 20 cm^−1^, indicating that the prepared MoS_2_ was a single layer [37]. SHG is an effective tool for revealing crystal symmetry. It is sensitive to the inversion symmetry of a crystal structure and can detect non-centrosymmetric materials with broken inversion symmetry. Monolayer MoS_2_ crystals produce strong SHG signals at an excitation wavelength of 1550 nm. Moreover, past research has shown that, after passing through the materials, two low-frequency light waves with a frequency of 193.4 THz can generate a dual-frequency light wave of 386.8 THz [38]. Figure 5d shows the peak of the SHG signal generated by the wide-bandgap MoS_2_ in our study at a 775 nm wavelength. The peak appears at half of the incident wavelength and the SHG intensity increases with increasing excitation power. Figure 5f shows a linear fit between the SHG intensity and excitation power for MoS_2_. The slope of the fit is 2.19 and is almost close to the theoretical value of 2. This indicates that the SHG intensity is quadratically related to the excitation power, which is consistent with the electric dipole theory [39]. In addition, the lattice symmetry of MoS_2_ was characterized in our study using polarization angle-dependent SHG intensity, as shown in Figure 5e [40]. For the vertical polarization component, the SHG intensity of MoS_2_ exhibited a clear sixfold rotational symmetry, which was consistent with its hexagonal structural symmetry. The fitting equation for second-harmonic intensity is $I=I_{0}{sin}^{2}(3\theta )$, where $I_{0}$ is the maximum value of SHG intensity.

### 4.2. Characterization of Wide-Bandgap MoS_2_ and Electrode on PET Surface

The monolayer MoS_2_ on the Si/SiO_2_ substrate was transferred to the PET substrate, and the electrodes were prepared, as shown in Figure 6. Figure 6a is the optical microscope image of triangular MoS_2_ with a size of about 50 μm on the PET flexible substrate, which shows that MoS_2_ has been completely transferred onto the PET substrate. Figure 6b is the comparison of the Raman spectra before and after transferring the MoS_2_. Compared to the former, the characteristic peak position and peak intensity of the Raman spectra of the sample transferred onto the PET do not change significantly. Figure 6c is the picture of the sensor. The sensor can be easily bent, which indicates that the flexibility of the sensor is high. Figure 6d is the optical microscopic image after lithography, and the electrode pattern in the channel location is pressed on both ends of the MoS_2_ along the armchair direction. Figure 6e,f are the optical microscopy and SEM images after the preparation of the electrode, which show that the lines of the metal electrode are straight and have no passivation. The electrodes are continuous and integral, without fracture or damage, and are, separately, dropped onto the monolayer MoS_2_ along the armchair direction.

### 4.3. Performance Testing and Discussion

The micro-displacement motor was set to step from small to large in one direction so that the probe at the front end of the micro-displacement motor push rod could compress the PET’s free end to generate the bending deformation, and the wide-bandgap MoS_2_ was subjected to the tensile strain accordingly. When the micro-displacement motor reached the set position, the corresponding peak current was recorded. Then, the push rod returned to the initial point, with the same stepping amount in turn, and the corresponding current value was recorded. The relationship curve between the stepping amount s and the current i was finally drawn.

Table 1 shows the relationship between the piezoelectric current of the wide-bandgap monolayer MoS_2_ and the stepping displacement of the stepper motor. Figure 7a is the schematic diagram of the structure of the monolayer MoS_2_-based piezoelectric sensor. The channel electrode is deposited along the armchair direction of the MoS_2_ on the surface of the PET substrate. Figure 7b shows the real-time piezoelectric performance test diagram of the sensor, in which the response current at 80 µm is negative, because, when step displacement is returned from 80 µm to the initial zero point, the piezoelectric potential of MoS_2_ along the armchair direction disappears, and the electrons in the external circuit appear to backflow. Figure 7c shows the relationship curve between the corresponding piezoelectric current and the stepping displacement. The stepping displacement of the stepper motor is approximately linear with the corresponding piezoelectric current. According to the beam structure theory and Saint-Venant’s small deformation theory, the stepping displacement of the stepper motor can be converted into the strain of monolayer MoS_2_ along the armchair direction. The relationship between the strain of the wide-bandgap monolayer MoS_2_ and the corresponding piezoelectric current was obtained by following the above. As shown in Figure 7d, the sensitivity of the device was calculated to be S = ∆I/∆ε ≈ 1.154 μA/%, according to the fitted curve.

## 5. Conclusions

In conclusion, this study successfully synthesized a large wide-bandgap monolayer MoS_2_ using the CVD technique, with an ammonium molybdate aqueous solution serving as the molybdenum source precursor. This approach enabled us to have precise control over the precursor quantity, facilitating the accurate growth of the monolayer MoS_2_. The subsequent transfer of this MoS_2_ layer onto a PET flexible substrate led to the successful fabrication of a micro-displacement sensor. SHG characterization confirmed the non-centrosymmetric crystal structure of the wide-bandgap MoS_2_, indirectly substantiating its piezoelectric properties. Performance testing of the sensor on a specialized platform revealed a piezoelectric response current of 5.12 nA for a 40 µm displacement of the PET substrate. Notably, within a displacement range of 40–100 µm, the sensor demonstrated an approximately linear relationship between the piezoelectric response current and the displacement, with a calculated sensitivity of 1.154 µA/%. The high sensitivity of this sensor to micro-displacements suggests its potential applicability in detecting and characterizing small surface fluctuations in objects, marking a significant advancement in the field of tactile sensing technology.

## Figures and Tables

**Figure 1 nanomaterials-14-00275-f001:**
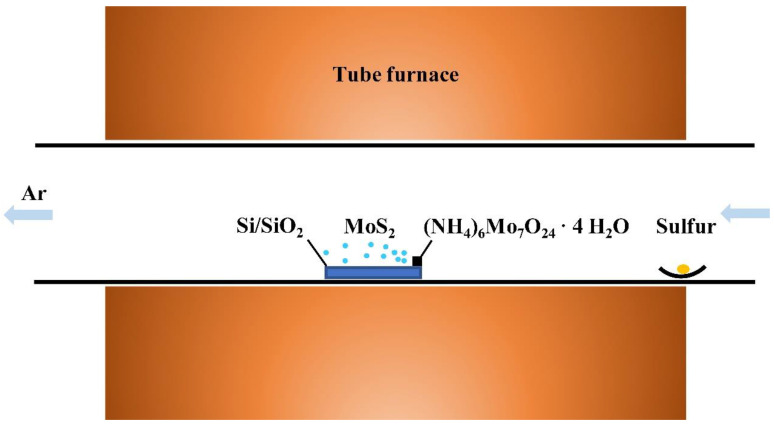
Sketch map of the CVD process for the growth of wide-bandgap monolayer MoS_2_.

**Figure 2 nanomaterials-14-00275-f002:**
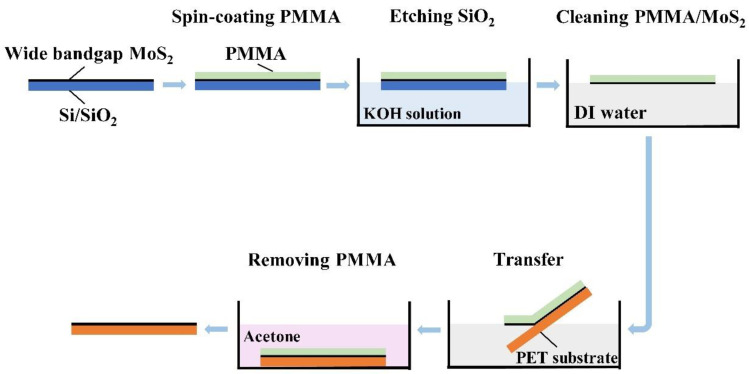
The flow chart of transferring wide-bandgap monolayer MoS_2_ using the wet-transfer method.

**Figure 3 nanomaterials-14-00275-f003:**
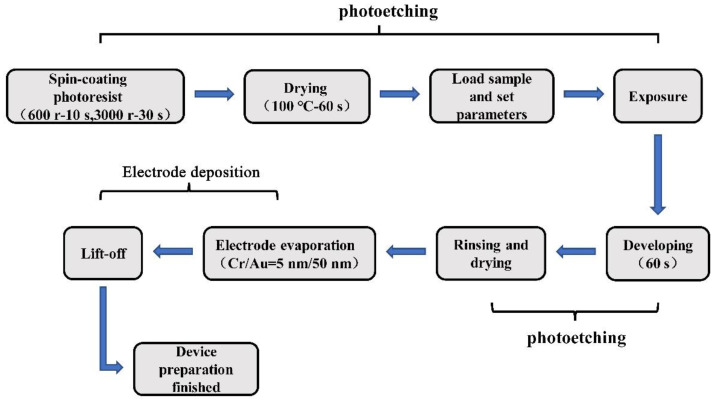
The flow chart of fabricating a wide-bandgap monolayer MoS_2_-based device.

**Figure 4 nanomaterials-14-00275-f004:**
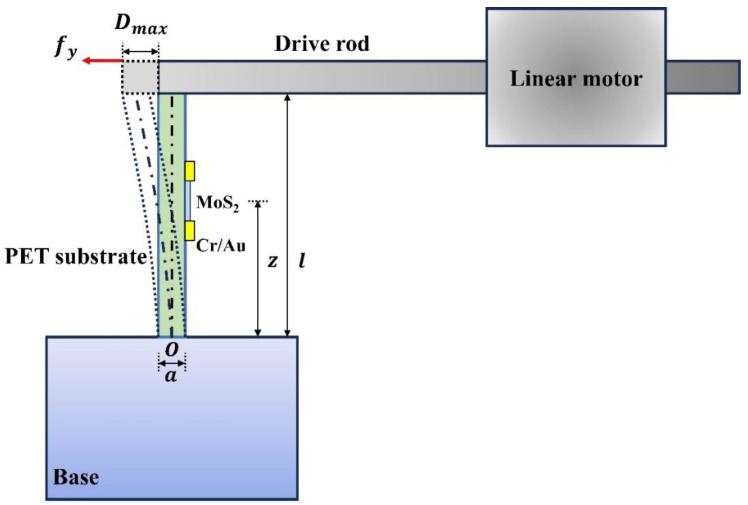
The schematic of the piezoelectric testing platform for a wide-bandgap monolayer MoS_2_-based device.

**Figure 5 nanomaterials-14-00275-f005:**
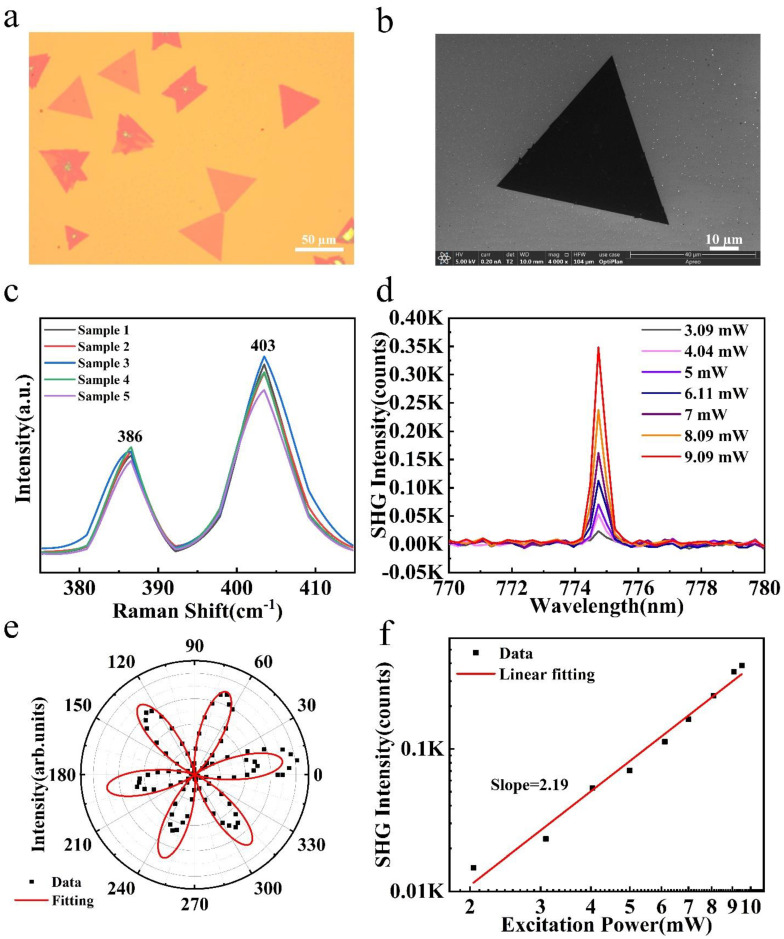
The characterization of wide-bandgap monolayer MoS_2_. (**a**) The optical image and (**b**) SEM characterization of the prepared MoS_2_ on a Si/SiO_2_ substrate. (**c**) Raman spectra of wide-bandgap monolayer MoS_2_. (**d**) Excitation power-dependent SHG intensity of monolayer MoS_2_. (**e**) Polar plot of polarization angle-dependent SHG intensity along the perpendicular direction of MoS_2_. (**f**) The linear fitting of SHG intensity versus excitation power—the coefficient is fitted to be 2.19.

**Figure 6 nanomaterials-14-00275-f006:**
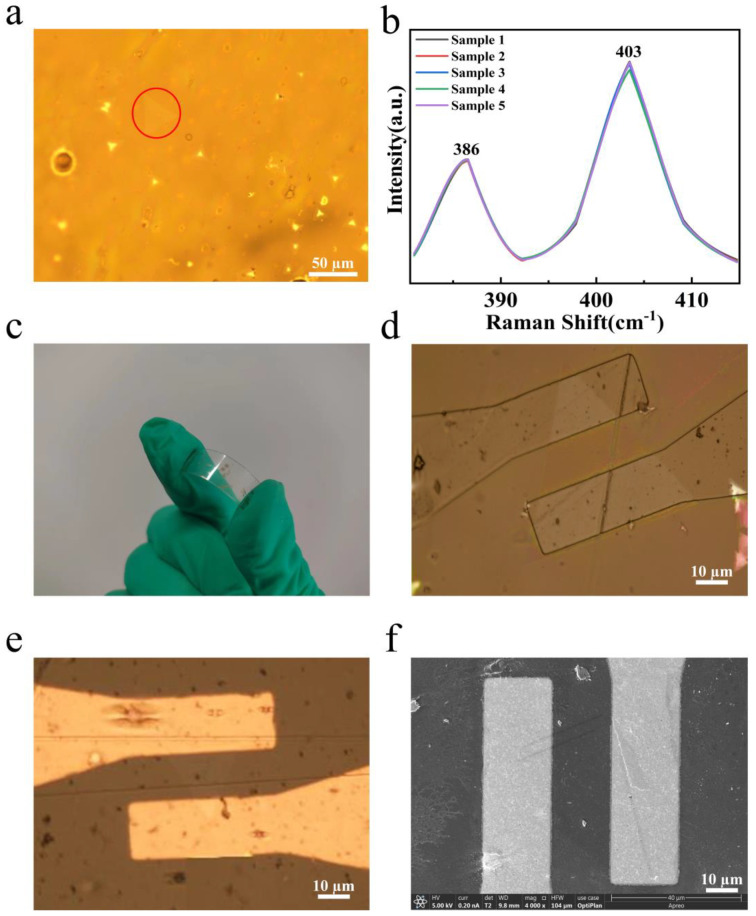
(**a**) The optical image and (**b**) Raman spectra of the wide-bandgap monolayer MoS_2_ transferred onto the PET substrate. The triangle MoS_2_ is marked with red circle in (**a**). (**c**) The picture of the sensor. The optical image of the pattern of the channel electrode after photolithography (**d**) and after thermal evaporation of the Cr/Au electrode. (**e**,**f**) SEM characterization of the prepared channel electrode.

**Figure 7 nanomaterials-14-00275-f007:**
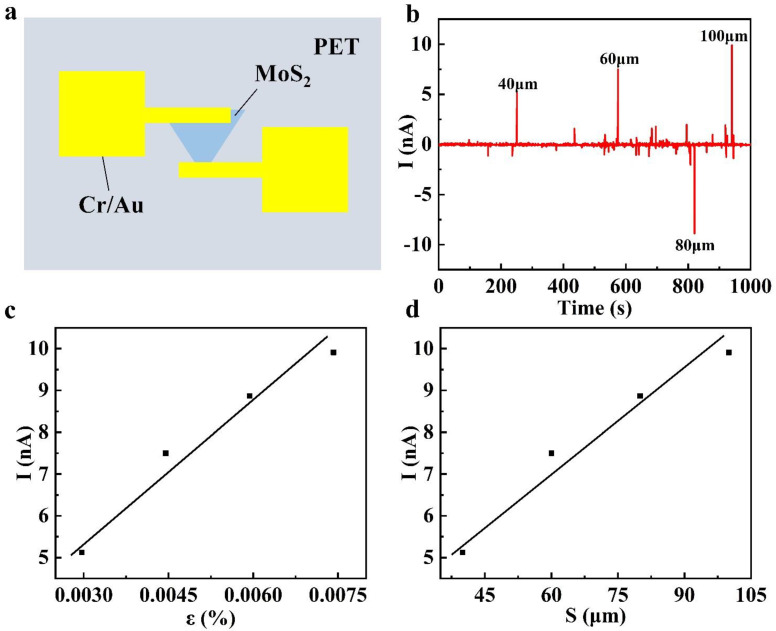
(**a**) Schematic diagram of the structure of the wide-bandgap monolayer MoS_2_-based piezoelectric sensor. (**b**) The real-time test diagram of the piezoelectric current under different displacements of the driving rod. (**c**) The relationship between the piezoelectric current and step displacement. (**d**) The relationship between the piezoelectric current and the strain applied in the armchair direction of the wide-bandgap monolayer MoS_2_.

**Table 1 nanomaterials-14-00275-t001:** The relationship between step displacement and the piezoelectric current.

Number	Step Displacement (μm)	Current I (nA)
1	40	5.12
2	60	7.50
3	80	8.87
4	100	9.91

## Data Availability

The data presented in this study are available on request from the corresponding author.

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
