# Peer review of "Growth of Wide-Bandgap Monolayer Molybdenum Disulfide for a Highly Sensitive Micro-Displacement Sensor"

_nanomaterials, 2024, doi:10.3390/nano14030275_

Round 1

Reviewer 1 Report

Comments and Suggestions for Authors

The topic is very interesting and timely. However, a few open questions should be answered prior to publication:
(1) Please specify "wide bandgap monolayer MoS2". (a) What is the band gap of the grown MoS2 crystals? (b) Do you have a direct proof that the triangular crystals are pure monolayer MoS2? For example, the optical image in Fig. 5 (a) shows triangles of different intensity. What is the origin of this difference? (see dark orange triangle in the upper right corner and moderate orange triangles left to this close to the bottom line). If a mixture of monolayer and bi- or trilayer MoS2 crystals are grown, how do you select the monolayer MoS2 for your sensor circuit?

(2) Figure 4 a shows that the MoS2 triangle on PET surface is clearly visible by means of optical contrast. In Fig. 4 d the triangle is visible in the area of the electrodes which is not covered by resist. Unfortunately, after metallization and lift off, no edges from the MoS2 crystal are visible anymore between the electrodes (Fig. 6e and f). Did you confirm that the MoS2 single crystal stayed intact? Please provide a proof for this.

(3) For the test procedure shown in Figure 4 it is important that the electrodes are arranged along the cantilever. In addition, they have to be put along the armchair directionof the MoS2 crystal. So, do you have to select a crystal that has by chance the correct orientation?

(4) A reference measurement for the clean PET cantilever should be given.  This can be done by repeating the measurement for a pure PET cantilever with electrode structure but w/o MoS2. By this means it can be made sure that there is clearly no contribution of the deflected PET cantilever to the tiny current response signal.

(5) The sensitivity values should be calculated from a statistical significant data set, e.g. an average of 50 to 100 data points per deflection value.

Technical issues:

- All micrographs and figures should be given with clear readable scale bars (Figs. 5a, 5b, 6a, 6d, 6e, 6f, 7a.

- Chinese characters in Fig. 3 should be replaced.

- The geometry of the electrodes in Fig. 6 is different from the schematic drawn in Fig. 7a. This is confusing and should be revised.

Comments on the Quality of English Language

Please revise the English Language with respect to grammar.
See for expample the sentences  "Compared with the typical molybdenum precursor..." (page 2/3 lines 77 - 80).

Reviewer 2 Report

Comments and Suggestions for Authors

The manuscript by WANG Shaopeng, et al. studies the growth of MoS2 single layers obtained by chemical vapour deposition. Moreover, the authors report on the piezoelectric properties of the grown materials and their potential application in displacement sensor. The topic could be of potential interest for the readers of the journal, I believe that several major revisions are necessary in order to make the manuscript suitable for publication. The points listed below claim for additional details and arguments so as to sustain the statements of the work.

Major comments:

11)  The title should be checked for grammar. What is the meaning of “highly micro-displacement sensor”? What does the adverb “highly” mean in the title?

22)  The manuscript lacks sufficient technical details to allow reproducibility of the results. Technical details of the Raman and Second Harmonic Generation techniques are missing.

33)  The sentence “Laser focusing and energy parameters were 60 and 0 respectively” is not scientifically meaningful. What do “60 and 0” values mean? What are the measured physical quantities?

44)  Error bars should be added to the data points. Statistical analysis of the data should be carried out considering the error bars.

55)  In the main text, the authors state that the slope resulting from the fit of the data in Figure 5 f is 1.94. On the contrary, in the figure it is reported 2.19. Please check.

66)  What is the theta angle introduced at page 8 row 200? Please, specify it in the text.

77)  The reader cannot appreciate the scale bars in SEM images.

88)  Figure 4 should be more informative, also a photo of the real system would be appreciated.

99)  The authors should comment further on the electrical performances of the device. How many devices did they test? To what extent the piezoelectric response is reproduced in the different devices? If possible, a comparison with similar devices based on other materials should be briefly presented in order to appreciate the advantages/disadvantages of the proposed devices.

Minor comments:

11)  The combination of words “wide bandgap” is repeated too many times in the text.

22)  In figure 3, please use only English words.

Comments on the Quality of English Language

English is fine. A few suggestions to improve the clarity of the text are reported in the review report

Round 2

Reviewer 1 Report

Comments and Suggestions for Authors

Thanks to the authors for their detailed ansers to the questions and for providing additional insightful material. The authors may consider to provide this material as supplementray information to all readers.

Reviewer 2 Report

Comments and Suggestions for Authors

The authors have made significant changes and improvements upon taking due care of all the questions/suggestions of this referee. The revised manuscript is now much more convincing and merits publication as it is

Comments on the Quality of English Language

English is fine